# Pseudocercospora rizhaoensis sp. nov. Causing Leaf Spot Disease of *Ligustrum japonicum* in China

**Yun Liu [1], Shumei Guo [1], Jin Liu [2] and Xiangli Yang [1,*]**

1    College of Agricultural Science and Technology, Shandong Agriculture and Engineering University, Jinan 250100, China

2    College of Forestry Engineering, Shandong Agriculture and Engineering University, Jinan 250100, China

*    Correspondence: z2013199@sdaeu.edu.cn

**Abstract:** *Ligustrum japonicum* is a common ornamental tree species in China. However, leaf spot disease has emerged in Rizhao City, Shandong Province of China in recent years. Members of *Pseudocercospora* are usually known as plant pathogens, mainly causing leaf spots and blights. Species of this genus are distinguished mainly based on morphological differences on the host plants, as well as the molecular data. A new species named *Pseudocercospora rizhaoensis* on *Ligustrum japonicum* is introduced herein based on morphology and molecular data of combined ITS, LSU, *act*, *tef1* and *rpb2* sequences. Koch's postulates were confirmed by a pathogenicity test, re-isolation and identification.

**Keywords:** morphology; Mycosphaerellacea; phylogeny; taxonomy

## 1. Introduction

*Pseudocercospora* Speg. (Mycosphaerellaceae, Mycosphaerellales) is a large cosmopolitan genus of plant pathogenic fungi that are commonly associated with leaf and fruit spots as well as blights on a wide range of plant hosts [1–3]. *Pseudocercospora* typed by *P. vitis* (Lév.) Speg. is distinguished from the other cercosporoid fungi by pale to dark olivaceous caespituli, pigmented conidia with unthickened and not refractive scars on the conidiogenous cells and hila at the basal ends of conidium in vivo [2,3].

Members of *Pseudocercospora* are distributed worldwide, but they are mostly abundant and diverse in tropical and subtropical areas and reproduce mainly by means of conidia [1,4–9]. Some species are associated with important plant diseases; for example, *Pseudocercospora fijiensis* (M. Morelet) Deighton is the causal agent of black Sigatoka leaf diseases of banana in Uganda and Tanzania [10]; *P. actinidiae* Deighton causes sooty spot disease on kiwifruit in Brazil [11]; *P. griseola* (Sacc.) Crous and U. Braun results in bean angular leaf spot disease in Ethiopia [12].

Species of *Pseudocercospora* are distinguished based on the morphology produced on the host plants and sequence data [1]. In addition, host information can also separate species of this genus, which is supported by the overall DNA phylogeny of ITS, LSU, *act* and *tef1* [1]. Subsequently, the *rpb2* locus was recommended to be added to the phylogeny for recognition of species within the genus *Pseudocercospora* [4]. *Ligustrum japonicum* Thunb. of the family Oleaceae Hoffmanns. and Link is native to central and southern Japan and Korea, and widely planted as an ornamental in parks and landscapes in China. During the surveys of plant diseases in Shandong Province, China, a colored cercosporoid fungus with fasciculate conidiophores, slightly thickened and darkened conidial scars and hilum was discovered, which causes a severe foliar disease on *Ligustrum japonicum*. Morphologically and phylogenetically, it was shown to be a species of *Pseudocercospora*. We compared its morphological features and molecular data to the known *Pseudocercospora* species and concluded that this species is new to science. Illustrations and detailed descriptions are provided for this new species herein.

## 2. Materials and Methods

### 2.1. Sample Survey, Collection and Fungal Isolation

Diseased leaf samples of *Ligustrum japonicum* were observed and collected in Rizhao City, Shandong Province of China (Figure 1), packed in paper bags and brought to the laboratory for isolation. The infected leaves were first surface-sterilized for 1 min in 75% ethanol, 3 min in 1.25% sodium hypochlorite, and 1 min in 75% ethanol, and then rinsed for 2 min in distilled water and blotted on dry sterile filter paper [13]. Then, the diseased areas of the samples were cut into 0.5 × 0.5 cm pieces using a double-edge blade, and transferred onto the surface of potato dextrose agar plates (PDA; 200 g potatoes, 20 g dextrose, 20 g agar per L) and incubated at 25 °C to obtain pure cultures. The cultures were deposited in the China Forestry Culture Collection Center (CFCC; http://cfcc.caf.ac.cn) and the specimen was deposited in the Herbarium of the Chinese Academy of Forestry (CAF; http://museum.caf.ac.cn).

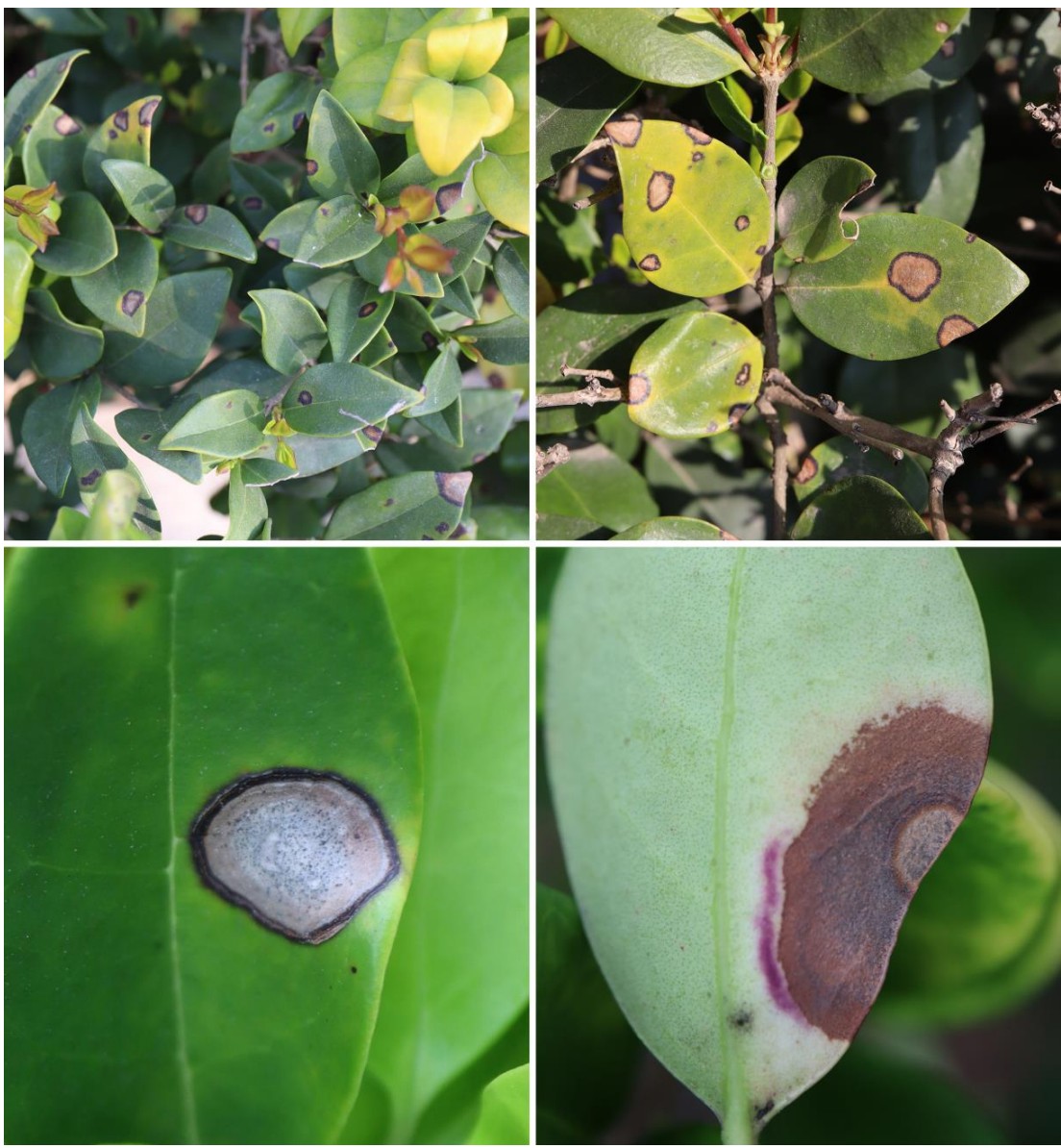

**Figure 1.** Symptoms of leaf spots on *Ligustrum japonicum*.

## 2.2. DNA Extraction, Sequencing and Phylogenetic Analyses

Genomic DNA was extracted from colonies grown on cellophane-covered PDA using a cetyltrimethylammonium bromide (CTAB) method [14]. DNA was checked by electrophoresis in 1% agarose gel, and the quality and quantity were measured using a NanoDrop 2000 (Thermo Scientific, Waltham, MA, USA). The nuclear ribosomal DNA internal transcribed spacers (ITS), large subunit rRNA (LSU), actin (*act*), translation elongation factor 1-alpha (*tef1*) and the second lar-gest RNA polymerase subunit (*rpb2*) regions were amplified using the primer pairs ITS4 (TCC TCC GCT TAT TGA TAT GC) and ITS5 (GGA AGT AAA AGT CGT AAC AAG G) [15], LROR (GTA CCC GCT GAA CTT AAG C) and LR5 (TCC TGA GGG AAA CTT CG) [16], ACT-512 (ATG TGC AAG GCC GGT TTC GC) and ACT-783 (TAC GAG TCC TTC TGG CCC AT) [17], EF1-668 (CAT CGA GAA GTT CGA GAA GG) and EF1-1251 (GGA RGT ACC AGT SAT CAT GTT) [17,18] and RPB2-5f2 (GGGGWGAYCAGAAGAAGGC) and fRPB2-7cR (CCCATRGCTTGYTTRCCCAT) [19], respectively. The PCR conditions were set as follows: an initial denaturation step of 5 min at 95 °C, followed by 35 cycles of denaturation at 94 °C for 1 min, 50 s at 52 °C (ITS and LSU) or 54 °C (*tef1*, *act* and *rpb2*) [20]. The final extension step was done at 72 °C for 7 min. The PCR products were examined by electrophoresis on 1.5% (*w/v*) agarose gels stained with ethidium bromide in 1 × TBE buffer. DNA sequencing was performed by the Shanghai Invitrogen Biological Technology Company Limited (Beijing, China).

DNA sequences were generated by using SeqMan v.7.1.0 from the DNASTAR Laser-gene software suite (DNASTAR Inc., Madison, WI, USA). Reference sequences used in the paper were downloaded from GenBank, and the GenBank accession numbers are listed in Table 1. *Trocophora simplex* was used as the outgroup taxon given its proposed relationship to *Pseudocercospora*. Sequences were aligned using MAFFT v.6 [21] and corrected manually using MEGA 7.0.21. The phylogenetic analyses of the combined loci were performed using the Maximum Likelihood (ML) and Bayesian Inference (BI) methods. ML was implemented on the CIPRES Science Gateway portal (https://www.phylo.org, accessed on 3 October 2022) using RAxML-HPC BlackBox 8.2.10 [22] employing a GTRGAMMA substitution model with 1000 bootstrap replicates. Bayesian inference was performed using a Markov Chain Monte Carlo (MCMC) algorithm in MrBayes v.3.2.6 [23]. The six simultaneous Markov chains were run for 1 M generations; starting from random trees and sampling trees every 100 th generation and 25% of aging samples were discarded, running until the average standard deviation of the split frequencies dropped below 0.01. The phylogram was visualized in FigTree v.1.3.1 (http://tree.bio.ed.ac.uk/software, accessed on 4 October 2022) and edited in Adobe Illustrator CS5 (Adobe Systems Inc., San Jose, CA, USA). The nucleotide sequence data of the new taxon were deposited in GenBank, and the GenBank accession numbers of all accessions included in the phylogenetic analyses are listed in Table 1.

## 2.3. Morphological Identification and Characterization

The morphology of the new species was studied based on the fruiting bodies formed on the diseased leaves. The fruiting bodies were observed and photographed under a dissecting microscope (M205 C, Leica, Wetzlar, Germany). The conidiogenous cells and conidia were immersed in tap water, and then the microscopic photographs were captured with an Axio Imager 2 microscope (Zeiss, Oberkochen, Germany) equipped with an Axiocam 506 color camera using Differential Interference Contrast (DIC) illumination. For measurements, more than 50 conidia were randomly selected. Culture characteristics were recorded from PDA after 20 d of incubation at 25 °C in the dark.

**Table 1.** Strains and GenBank accession numbers used in this study.

| Species | Isolates | GenBank Accession Numbers | | | | |
|---|---|---|---|---|---|---|
| | | LSU | ITS | *act* | *tef1* | *rpb2* |
| *Pseudocercospora abeliae* | MUCC1674 * | NA | LC599330 | LC599407 | LC599448 | LC599587 |
| *P. aeschynomenicola* | CPC 25227 = COAD 1972 * | KT290173 | KT290146 | KT313501 | KT290200 | NA |
| *P. airliensis* | BRIP 58550 * | KM055433 | KM055429 | NA | KM055436 | NA |
| *P. aleuritis* | MAFF237174 = MUCC1230 * | NA | LC599331 | LC599408 | LC599449 | LC599588 |
| *P. amelanchieris* | MAFF 237782 = MUCC885 * | NA | KX462583 | KX462550 | KX462669 | KX462616 |
| *P. ampelopsis* | CBS 131583 = CPC 11680 * | GU253846 | GU269830 | GU320534 | GU384542 | NA |
| *P. angiopteridis* | CBS 147385 * | NA | LC599332 | LC599409 | LC599450 | LC599589 |
| *P. angolensis* | CBS 149.53 * | JQ324941 | JQ324975 | JQ325011 | JQ324988 | NA |
| *P. araliae* | MUCC 873 * | GU253702 | GU269653 | GU320361 | GU384371 | KX462617 |
| *P. arecacearum* | CBS 118406 * | GU253704 | GU269655 | GU320363 | GU384373 | NA |
| *P. assamensis* | CBS 122467 * | GU253705 | GU269656 | GU320364 | GU384374 | NA |
| *P. avicenniae* | CBS 146479 * | NA | GU188047 | LC599410 | LC599451 | LC599590 |
| *P. basiramifera* | CBS 111072 = CPC 1266 * | GU253709 | GU269661 | GU320368 | DQ211677 | NA |
| *P. basitruncata* | CBS 114664 = CPC 1202 * | GU253710 | DQ267600 | DQ147622 | DQ211675 | NA |
| *P. biophyti* | CPC 20020 | NA | LC599333 | LC599411 | LC599452 | LC599591 |
| *P. bixae* | CPC 25244 = COAD 1563 * | KT290180 | KT290153 | KT313508 | KT290207 | NA |
| *P. brackenicola* | CPC 24695 = COAD 1991 * | KT037565 | KT037524 | KT037606 | KT037484 | NA |
| *P. breonadiae* | CBS 143489 = CPC 30153 * | MH107959 | MH107913 | MH107985 | MH108026 | MH108006 |
| *P. bruceae* | MUCC 2875 * | NA | LC599334 | LC599412 | LC599453 | NA |
| *P. casuarinae* | CBS 128218 * | HQ599604 | HQ599603 | LC599413 | LC599454 | NA |
| *P. ceratoniae* | CBS 147386 * | NA | LC599335 | LC599414 | LC599455 | LC599592 |
| *P. cercidicola* | MAFF 237791 = MUCC 896 * | GU253719 | GU269671 | GU320377 | GU384388 | KX462618 |
| *P. cercidis-chinensis* | CBS 132109 = CPC 14481 * | JX901884 | GU269670 | GU320376 | GU384387 | LC599593 |
| *P. chamaecristae* | CPC 25228 = COAD 1973 * | KT290174 | KT290147 | KT313502 | KT290201 | NA |
| *P. chiangmaiensis* | CBS 123244 * | NG042738 | EU882113 | KF903544 | KF903177 | NA |
| *P. chibaensis* | MUCC1670E * | NA | KX462584 | KX462551 | KX462670 | KX462619 |
| *P. chionanthi-retusi* | TUA50 = NCHUPP L1605 * | NA | KX462585 | KX462552 | KX462671 | KX462620 |
| *P. cladrastidis* | MUCC1494 * | NA | LC599336 | LC599415 | LC599457 | LC599594 |
| *P. convoluta* | CBS 113377 * | MF951226 | DQ676519 | NA | NA | MF951617 |
| *P. coprosmae* | CBS 114639 * | JQ324946 | GU269680 | GU320386 | GU384397 | NA |
| *P. cordiana* | CPC 2552 * | GU214472 | AF362054 | GU320387 | GU384398 | NA |
| *P. corylopsidis* | MAFF 237795 = MUCC 908 * | NG069064 | GU269684 | GU320390 | GU384401 | KX462621 |
| *P. cotini* | MAFF410088 = MUCC1415 * | NA | LC599337 | LC599416 | LC599458 | LC599596 |
| *P. cotoneastri* | MAFF 410089 = MUCC1416 * | NA | KX462586 | KX462553 | KX462672 | KX462622 |

**Table 1.** *Cont.*

| Species | Isolates | GenBank Accession Numbers | | | | |
|---|---|---|---|---|---|---|
| | | **LSU** | **ITS** | *act* | *tef1* | *rpb2* |
| *P. crispans* | CBS 125999 = CPC 14883 * | GU253825 | GU269807 | GU320510 | GU384518 | KX462623 |
| *P. crocea* | CBS 126004 = CPC 11668 * | JQ324947 | GU269792 | GU320493 | GU384502 | NA |
| *P. crousii* | CBS 119487 | GQ852631 | GU269686 | GU320392 | GU384403 | NA |
| *P. cryptomeriicola* | MAFF240073 = NBRC 102150 * | NA | LC599338 | LC599418 | LC599460 | LC599598 |
| *P. curcumicola* | MUCC733 * | NA | LC599339 | LC599419 | LC599461 | LC599599 |
| *P. cyathicola* | CBS 129520 = CPC 17047 * | JF951159 | JF951139 | KX462554 | KX462673 | KX462624 |
| *P. cymbidiicola* | CBS 115132 * | GU253733 | GU269692 | GU320397 | GU384408 | NA |
| *P. dalbergiae* | TUA55 * | NA | LC599340 | LC599420 | LC599462 | LC599600 |
| *P. daphniphylli* | MAFF 410009 = MUCC1399 * | NA | KX462587 | KX462555 | KX462674 | KX462625 |
| *P. davidiicola* | MAFF 240281 = MUCC296 * | GU253734 | GU269693 | GU320398 | GU384409 | KX462626 |
| *P. delonicicola* | MUCC2869 * | NA | LC599341 | LC599421 | LC599463 | LC599601 |
| *P. dingleyae* | CBS 114645 * | KX286997 | KX287299 | NA | NA | KX288454 |
| *P. diplusodonii* | CPC 25179 = COAD 1476 * | KT290162 | KT290135 | KT313490 | KT290189 | NA |
| *P. dodonaeae* | CBS 114647 * | JQ324948 | GU269697 | JQ325013 | GU384413 | NA |
| *P. dovyalidis* | CBS 126002 = CPC 13771 * | GU253818 | GU269800 | GU320503 | GU384513 | NA |
| *P. ebulicola* | CBS 147387 * | NA | LC599342 | LC599422 | NA | NA |
| *P. elaeocarpicola* | MAFF 237189 = MUCC1236 * | NA | KX462588 | KX462556 | KX462675 | KX462627 |
| *P. emmoticola* | CPC 25187 = COAD 1491 * | KT290163 | KT290136 | KT313491 | KT290190 | NA |
| *P. eriobotryae* | MUCC 1007 * | NA | KX462589 | KX462557 | KX462676 | KX462628 |
| *P. eriobotryicola* | TUA12 = NCHUPPL1601 * | NA | KX462590 | KX462558 | KX462677 | KX462629 |
| *P. ershadii* | CBS 136114 = CCTU 1206 * | KP717032 | KM452867 | KM452844 | KM452889 | MN786459 |
| *P. eucalyptorum* | CBS 114866 = CPC 11 * | JQ739817 | KF901720 | KF903474 | KF903195 | MF951618 |
| *P. eumusae* | CBS 114824 * | NA | EU514238 | NA | NA | NA |
| *P. euonymi-japonici* | CGMCC 3.18576 * | NA | MH255812 | NA | NA | MH392531 |
| *P. eupatoriella* | CBS 113372 * | GU253743 | GU269704 | GU320408 | GU384420 | NA |
| *P. eupatorii- formosani* | TUA59 = NCHUPP L1606 * | NA | KX462591 | KX462559 | KX462678 | KX462630 |
| *P. euphorbiacearum* | COAD 1537 * | KT290172 | KT290145 | KT313500 | KT290199 | NA |
| *P. exilis* | CPC 25193 = COAD 1501 * | KT290166 | KT290139 | KT313494 | KT290193 | NA |
| *P. farfugii* | MUCC978 * | NA | LC599343 | LC599423 | LC599464 | LC599603 |
| *P. fijiensis* | CBS 120258 = CIRAD 86 * | JQ324952 | EU514248 | NA | NA | NA |
| *P. flavomarginata* | CBS 126001 * | NA | GU269804 | GU320507 | GU384515 | LC599604 |
| *P. fori* | CBS 113285 * | NA | AF468869 | KF903462 | NA | KT356874 |
| *P. formosana* | MUCC2612 * | NA | LC599344 | LC599424 | LC599466 | LC599605 |
| *P. forsythiae* | MAFF 410087 = MUCC1414 * | NA | LC599345 | LC599425 | LC599467 | NA |

Table 1. *Cont.*

| Species | Isolates | GenBank Accession Numbers | | | | |
|---|---|---|---|---|---|---|
| | | LSU | ITS | *act* | *tef1* | *rpb2* |
| *P. fukuii* | MAFF238121 = MUCC1297 * | NA | LC599347 | LC599427 | LC599469 | LC599607 |
| *P. fukuokaensis* | MAFF 237768 = MUCC 887 * | GU253751 | GU269714 | GU320418 | GU384430 | KX462632 |
| *P. ginkgoana* | R. Kirschner 3563 (TNM) * | NA | JX134048 | NA | NA | NA |
| *P. glochidionis* | MAFF 237000; MUCC1211 * | NA | LC599348 | LC599428 | LC599470 | LC599608 |
| *P. gracilis* | CBS 242.94 * | NA | DQ267582 | NA | DQ211666 | NA |
| *P. griseola f. griseola* | CBS 119906 * | NA | DQ289812. | NA | NA | NA |
| *P. griseola f. mesoamericana* | CBS 119113 * | NA | DQ289824 | NA | NA | NA |
| *P. hachijokibushii* | MAFF 238479 * | NA | KX462593 | KX462561 | KX462680 | KX462633 |
| *P. haiweiensis* | CBS 131584 = CPC 14084 * | GU253821 | GU269803 | GU320506 | GU384514 | KX462634 |
| *P. hardenbergiae* | CBS 147381 * | NA | LC599349 | LC599429 | LC599471 | LC599609 |
| *P. heteropyxidicola* | CBS 146082 = CPC 38030 * | NA | MN562151 | MN556791 | NA | NA |
| *P. hiratsukana* | MAFF 238300 = MUCC1105 * | NA | KX462594 | KX462562 | KX462681 | KX462635 |
| *P. houttuyniae* | MAFF 238071 = MUCC1289 * | NA | KX462595 | KX462563 | KX462682 | KX462636 |
| *P. humuli* | MUCC 742 * | GU253758 | GU269725 | GU320428 | GU384439 | KX462637 |
| *P. humulicola* | CBS 131585 * | JQ324956 | GU269723 | GU320427 | GU384438 | NA |
| *P. imazekii* | MUCC 1668 * | NA | KX462596 | KX462564 | KX462683 | KX462638 |
| *P. indonesiana* | CBS 122473 * | NA | GU269735 | GU320437 | GU384448 | NA |
| *P. iwakiensis* | MUCC 1736 * | NA | KX462607 | KX462574 | KX462693 | KX462657 |
| *P. ixoriana* | MUCC2608 * | NA | LC599350 | LC599430 | LC599472 | LC599610 |
| *P. izuohshimense* | MAFF 238478 = MUCC1336 * | NA | KX462597 | KX462565 | KX462684 | KX462639 |
| *P. jagerae* | BRIP 58549 * | NA | KM055431 | NA | KM055438 | NA |
| *P. kadsurae* | MUCC 752 * | NA | KX462598 | KX462566 | KX462685 | KX462640 |
| *P. kaki* | MAFF 238214 * | GU253761 | LC512001 | LC512007 | LC515783 | LC515794 |
| *P. kakiicola* | MAFF 238238 = MUCC 900 * | NA | GU269729 | GU320431 | GU384442 | NA |
| *P. kenyirana* | MUCC 2873 * | NA | LC599351 | LC599431 | LC599473 | NA |
| *P. kiggelariae* | CBS 132016 = CPC 11853 * | GU253762 | GU269730 | GU320432 | GU384443 | NA |
| *P. kobayashiana* | MAFF 236999 * | NA | LC511998 | LC512004 | LC515780 | LC515791 |
| *P. leandrae-fragilis* | COAD 1977 * | NA | KY574288 | NA | NA | NA |
| *P. leucadendri* | CPC 1869 * | GU214480 | GU269842 | GU320545 | GU384555 | NA |
| *P. liquidambaricola* | MAFF410455 * | NA | LC599352 | LC599432 | LC599474 | LC599611 |
| *P. longispora* | CBS 122470 * | NA | GU269734 | GU320436 | GU384447 | NA |

Table 1. *Cont.*

| Species | Isolates | GenBank Accession Numbers | | | | |
|---|---|---|---|---|---|---|
| | | **LSU** | **ITS** | *act* | *tef1* | *rpb2* |
| *P. lonicericola* | MUCC 889 = MAFF 237785 * | GU253766 | GU269736 | GU320438 | JQ324999 | KX462641 |
| *P. luzardii* | CPC 25196 = COAD 1505 * | KT290167 | KT290140 | KT313495 | KT290194 | NA |
| *P. lyoniae* | MAFF 237775 = MUCC 910 * | GU253768 | GU269739 | GU320441 | GU384451 | KX462642 |
| *P. lythri* | CBS 132115 = CPC 14588 * | NA | GU269742 | GU320444 | GU384454 | LC599612 |
| *P. macadamiae* | CBS 133432 * | KX286998 | KX287300 | KU878551 | KU878504 | KX288455 |
| *P. macrospora* | CBS 114696 = CPC 25538 | GU214478 | AF362055 | GU320447 | GU384457 | NA |
| *P. madagascariensis* | CBS 124155 * | NA | GQ852767 | KF253625 | KF253265 | KX462643 |
| *P. maetaengensis* | MFLUCC 14-0411 * | NA | MN648323 | NA | NA | NA |
| *P. mangifericola* | BRIP 52776b * | NA | GU188048 | NA | NA | NA |
| *P. manihotii* | CPC 25219 = COAD 1534 * | KT290171 | KT290144 | KT313499 | KT290198 | NA |
| *P. mapelanensis* | CMW40581 * | KM203121 | KM203118 | KM203127 | KM203124 | NA |
| *P. marginalis* | CBS 131582 = CPC 12497 * | GU253812 | GU269794 | GU320495 | GU384504 | NA |
| *P. mazandaranensis* | CCTU 1102 = CBS 136115 * | KP717020 | KM452854 | KM452831 | KM452876 | LC599613 |
| *P. metrosideri* | CBS 114294 * | KX286999 | KX287301 | NA | NA | KX288456 |
| *P. microlepiae* | BCRC FU30353 * | NA | KR348740 | NA | NA | NA |
| *P. musae* | CBS 116634 * | GU253775 | GU269747 | GU320449 | GU384459 | NA |
| *P. naitoi* | MAFF 237906 = MUCC1072 * | NA | KX462599 | KX462567 | KX462686 | KX462644 |
| *P. nandinae* | MAFF 237633 = MUCC1260 * | NA | KX462600 | KX462568 | KX462687 | KX462645 |
| *P. natalensis* | CBS 111069 = CPC 1263 * | DQ267576 | DQ303077 | DQ147620 | JQ325000 | NA |
| *P. nelumbonicola* | BCRC FU30367 * | NA | KY304492 | NA | NA | NA |
| *P. neriicola* | CBS 138010 = CPC 23765 * | KJ869222 | KJ869165 | KJ869231 | KJ869240 | KX462647 |
| *P. nodosa* | CBS 554.71 * | MF951227 | MF951367 | NA | NA | MF951620 |
| *P. norchiensis* | CBS 120738 = CPC 13049 * | GU253780 | EF394859 | GU320455 | GU384464 | KX462648 |
| *P. ocimi-basilici* | CPC 10283 * | NA | GU269754 | GU320456 | GU384465 | NA |
| *P. paederiae* | MAFF 239161 | NA | KX462603 | KX462570 | KX462689 | KX462651 |
| *P. palleobrunnea* | CBS 124771 = CPC 13387 * | GQ303319 | GQ303288 | GU320500 | GU384509 | KX462652 |
| *P. pancratii* | CBS 137.94 * | GU253784 | GU269759 | GU320460 | GU384470 | NA |
| *P. paranaensis* | CPC 24680 = COAD 1987T | KT037563 | KT037522 | KT037604 | KT037482 | NA |
| *P. parapseudarthriae* | CBS 137996 = CPC 23449 * | KJ869208 | KJ869151 | KJ869229 | KJ869238 | NA |
| *P. perae* | CPC 25171 = COAD 1465 * | KT290159 | KT290132 | KT313487 | KT290186 | NA |
| *P. perrottetiae* | CBS 147382 * | NA | LC599353 | LC599433 | LC599477 | LC599614 |
| *P. photiniae* | MUCC 1661 * | NA | KX462604 | KX462571 | KX462690 | KX462653 |
| *P. pini-densiflorae* | MUCC 1714 * | NA | LC599354 | LC599434 | LC599478 | LC599615 |

**Table 1.** *Cont.*

| Species | Isolates | GenBank Accession Numbers | | | | |
|---|---|---|---|---|---|---|
| | | **LSU** | **ITS** | *act* | *tef1* | *rpb2* |
| *P. planaltinensis* | CPC 25189 = COAD 1495 * | KT290164 | KT290137 | KT313492 | KT290191 | NA |
| *P. platyceriicola* | MUCC2876 * | NA | LC599355 | LC599435 | LC599479 | LC599616 |
| *P. plectranthi* | CBS 131586 = CPC 11462 * | NG070621 | GU269791 | GU320492 | GU384501 | NA |
| *P. plumeriifolii* | CPC 25191 = COAD 1498 * | KT290165 | KT290138 | KT313493 | KT290192 | NA |
| *P. pothomorphes* | CPC 25166 = COAD 1450 * | KT290158 | KT290131 | KT313486 | KT290185 | NA |
| *P. profusa* | CBS 132306 = CPC 10055 * | GU253787 | GU269762 | GU320463 | GU384473 | NA |
| *P. proiphydis* | BRIP 58545 * | KM055434 | KM055430 | NA | KM055437 | NA |
| *P. proteae* | CBS 131587 = CPC 15217 * | MH877381 | GU269808 | GU320511 | GU384519/ | NA |
| *P. pruni-grayanae* | MUCC 1715 * | NA | LC599356 | NA | LC599481 | LC599618 |
| *P. pseudomusae* | CBS 147147 * | NA | MW063423 | MW070772 | MW071091 | MW070919 |
| *P. pseudomyrticola* | CBS 145554 = CPC 35448 * | MK876446 | MK876405 | MK876461 | MK876499 | MK876490 |
| *P. pseudostigminaplatani* | CBS 131588 = CPC 11726 * | JQ324963 | GU269857 | GU320560 | GU384568 | NA |
| *P. punctata* | CBS 132116 = CPC 14734 * | GU253791 | GU269765 | GU320468 | GU384477 | MF951622 |
| *P. punicae* | MAFF236998 = MUCC 1209 | NA | KX462606 | KX462573 | KX462692 | KX462655 |
| *P. pyracanthae* | MAFF237140 = MUCC 1226 * | GU253792 | GU269767 | NA | GU384479 | LC599619 |
| *P. pyracanthigena* | CBS 131589 = CPC 10808 * | NA | GU269766 | GU320469 | GU384478 | NA |
| *P. ravenalicola* | CBS 122468 * | GU253828 | GU269810 | GU320513 | GU384521 | NA |
| *P. rhabdothamni* | CBS 114872 * | JQ324964 | GU269768 | GU320471 | GU384480 | NA |
| *P. rhamnellae* | CBS 131590 = CPC 12500 * | GU253813 | GU269795 | GU320496 | GU384505 | NA |
| *P. rhapisicola* | MAFF305042 = MUCC1484 * | NA | LC599357 | LC599436 | LC599483 | LC599620 |
| *P. rhododendri-indici* | CBS 131591 = CPC 10822 * | JQ324965 | GU269722 | GU320426 | NA | NA |
| *P. riachueli* var. *horiana* | MUCC2141 * | NA | LC599358 | LC599437 | LC599484 | LC599621 |
| *P. richardsoniicola* | CPC 25248 = COAD 1568 * | KT290181 | KT290154 | KT313509 | KT290208 | NA |
| *P. rigidae* | CPC 25175 = COAD 1472 * | KT290161 | KT290134 | KT313489 | KT290188 | NA |
| *P. rosae* | MFLUCC 14-0408 * | MG829063 | MG828952 | NA | NA | NA |
| **P. rizhaoensis** | **CFCC 57581 *** | **NA** | **OP661350** | **OP651770** | **OP651772** | **OP651774** |
| **P. rizhaoensis** | **CFCC 57582** | **NA** | **OP661351** | **OP651771** | **OP651773** | **OP651775** |
| *P. sambucigena* | CBS 126000 * | GU253809 | GU269788 | GU320508 | GU384498 | NA |
| *P. sawadae* | MAFF 239714 | NA | LC599359 | LC599438 | LC599485 | LC599622 |
| *P. schizolobii* | CBS 120029 = CPC 12962 * | KF251826 | KF251322 | KF253628 | KF253269 | NA |
| *P. sennae-multijugae* | CPC 25206 = COAD 1519 * | KT290169 | KT290142 | KT313497 | KT290196 | NA |
| *P. serpocaulonicola* | CPC 25077 = COAD 1866 * | KT037566 | KT037525 | KT037607 | KT037485 | NA |

**Table 1.** *Cont.*

| Species | Isolates | GenBank Accession Numbers | | | | |
|---------|----------|------|-----|-----|------|------|
| | | **LSU** | **ITS** | *act* | *tef1* | *rpb2* |
| *P. solani-pseudocapsicicola* | CPC 25229 = COAD 1974 * | KT290175 | KT290148 | KT313503 | KT290202 | NA |
| *P. sophoricola* | CCTU 1037 = CBS 136020 * | KP717027 | KM452861 | KM452838 | KM452883 | MW272931 |
| *P. sphaerulinae* | CBS 112621 * | KF901958 | KF901625 | NA | KF903215 | NA |
| *P. stemonicola* | MUCC2874 * | NA | LC599360 | LC599439 | LC599487 | NA |
| *P. stephanandrae* | MAFF237799 = MUCC914 * | GU253831 | GU269814 | GU320516 | GU384526 | KX462658 |
| *P. stranvaesiae* | MAFF410090 = MUCC1417 * | NA | LC599361 | LC599440 | LC599488 | LC599623 |
| *P. struthanthi* | CPC 25199 = COAD 1512 * | KT290168 | KT290141 | KT313496 | KT290195 | NA |
| *P. styracina* | COAD 2369 * | MH480643 | MH397664 | MH480641 | MH480642 | NA |
| *P. symploci* | NCHUPP L1685 = CBS142471 * | NA | LC599362 | LC599441 | LC599489 | LC599624 |
| *P. tabernaemontanae* | CPC 19198 * | NA | LC599363 | LC599442 | NA | LC599625 |
| *P. tereticornis* | CBS 125214 = CPC 13299 * | NA | GQ852770 | GU320499 | GU384508 | KX462659 |
| *P. terengganuensis* | MUCC2871 * | NA | LC599364 | LC599443 | LC599490 | NA |
| *P. tinea* | TUA40 = NCHUPP L1603 * | NA | KX462608 | KX462577 | KX462696 | KX462660 |
| *P. togashiana* | MAFF410006 * | NA | LC599365 | LC599444 | LC599491 | LC599626 |
| *P. trichogena* | CPC 24664 = COAD 1087 * | KT037560 | KT037519 | KT037601 | KT037479 | NA |
| *P. trinidadensis* | COAD 1756 * | NA | KT290157 | NA | KT290210 | NA |
| *P. tumulosa* | CBS 121158 * | NA | DQ530217 | NA | NA | NA |
| *P. vassobiae* | CPC 25251 = COAD 1572 * | KT290182 | KT290155 | KT313510 | NA | NA |
| *P. viburnigena* | CBS 125998 = CPC 15249 * | GU253827 | GU269809 | GU320512 | GU320512 | NA |
| *P. violamaculans* | MUCC 1660 * | NA | KX462610 | KX462579 | KX462698 | KX462662 |
| *P. vitis* | CBS 132012 =CPC 11595 | GU214483 | GU269829 | GU320533 | GU384541 | KX462663 |
| *P. wulffiae* | CPC 25232 = COAD 1976 * | KT290177 | KT290150 | KT313505 | KT290204 | NA |
| *P. xanthocercidis* | CBS 131593 = CPC 11665Iso * | JQ324971 | JQ324983 | JQ325026 | JQ325005 | NA |
| *P. xenopunicae* | CBS 147384 * | NA | LC599367 | LC599446 | LC599493 | LC599628 |
| *P. xenosyzygiicola* | MAFF237986 = MUCC1481 * | NA | KX462611 | KX462580 | KX462699 | KX462664 |
| *P. xylopiae* | CPC 25173 = COAD 1469 * | KT290160 | KT290133 | KT313488 | KT290187 | NA |
| *P. yakushimensis* | MAFF237025 = MUCC1214 * | NA | LC599368 | LC599447 | LC599494 | LC599629 |
| *P. zambiae* | CBS 136423 = CPC 22686 * | NA | KF777175 | NA | NA | MF951630 |
| *P. zelkovae* | MAFF 238237 = MUCC872 * | NA | GU269835 | GU320537 | GU384547 | KX462665 |
| *Trochophora simplex* | CBS 124744 | NA | NA | GU320568 | GU384580 | KX462666 |

Note: NA, not applicable. Ex-type strains are marked with *, and strains from the present study are in black bold.

### 2.4. Pathogenicity Testing

Two isolates of *Pseudocercospora rizhaoensis* (ex-type strain: CFCC 57581; CFCC 52288) were used for inoculations, and agar plugs were used as the negative control. Detached healthy *Ligustrum japonicum* leaves were used for artificial inoculation experiments. The leaves were surface-sterilized with 75% ethanol, rinsing three times in sterile water, and then we waited for the surface moisture to dry. Discs of agar were cut from the actively growing margins of the cultures and these were placed on the non-wounded in vitro leaves. Each group was performed three times and cultured at 25 °C in the dark.

## 3. Results

### 3.1. Phylogeny

The combined sequence dataset (ITS, LSU, *tef1*, *act* and *rpb2*) was analyzed to infer the phylogenetic placement of our new isolates within the genus *Pseudocercospora*. The dataset consisted of 193 sequences, including an outgroup taxon, *Trocophora simplex* (CBS124744). A total of 2755 characters, including gaps (510 for ITS, 788 for LSU, 543 for *tef1*, 241 for *act*, and 673 for *rpb2*), were included in the phylogenetic analysis. The best ML tree (lnL = − 33,732.84) revealed by RA×ML is shown in Figure 2. The topologies resulting from ML and BI analyses of the concatenated dataset were congruent (Figure 2). The phylogenetic tree showed that isolates CFCC 57581 and CFCC 57582 from the present study formed into a distinguished clade from the other known *Pseudocercospora* species.

### 3.2. Taxonomy

*Pseudocercospora rizhaoensis* Yun Liu, sp. nov. Figure 3.

Mycobank no.: 845997.

Etymology—Named after Rizhao City, where the holotype was collected.

Description—*Leaf spots* amphigenous, circular, scattered, pale brown to brown with reddish brown margin. *Caespituli* hypophyllous, synnematous with blackish brown conidiophores. *Mycelium* internal, hyaline to brown. *Stromata* hypophyllous, substomatal, epidermal, erumpent, well-developed, subglobose to globose, dark brown to blackish, 25–85 μm diam. *Conidiophores* dense, arising from the upper part of stromata, straight to sinuous-geniculated, cylindrical, unbranched, pale brown to brown, paler towards the apex, 4.5–25 × 2–3.5 μm, 0–2-septate, smooth. *Conidiogenous cells* integrated, terminal, proliferating percurrently or sympodially, with unthickened and truncated conidial loci. *Conidia* solitary, holoblastic, cylindrical to obclavate, 15–50.5 × 2.5–4.5 μm, 1–5-septate, hyaline- to pale-colored, smooth, acute at the apex, obconically truncated, unthickened and not darkened at the base.

Culture characteristics—*Colonies* on PDA flat, spreading, with flocculent aerial mycelium, edge entire, mouse grey, reaching 60 mm diam. after 20 d at 25 °C.

Material examined—CHINA, Shandong Province, Rizhao City, Beijinglu Street, on diseased leaves of *Ligustrum japonicum*, 7 August 2021, Yun Liu (CAF800066 *holotype*; *ex-type living culture*, CFCC 57581); Ibid. Shandong Province, Rizhao City, Shijiu Street, on diseased leaves of *Ligustrum japonicum*, 12 August 2021, Yun Liu (culture CFCC 57582).

Notes—Two isolates of *Pseudocercospora* from leaf spots of *Ligustrum japonicum* clustered into a well-supported clade distinguished from the other members within this genus (Figure 2), which is proposed as *P. rizhaoensis* herein. Phylogenetically, *P. rizhaoensis* is close to *P. eupatoriella* Crous and Den Breeÿen and *P. ginkgoana* R. Kirschner; however, these species can be distinguished by their hosts (*P. rizhaoensis* on *Ligustrum japonicum* vs. *P. eupatoriella* on *Chromolaena odorata* R. M. King and H. Robinson vs. *P. ginkgoana* on *Ginkgo biloba* L.) [1,24]. Morphologically, conidia of *P. rizhaoensis* is similar to those of *P. eupatoriella*, but wider than those of *P. ginkgoana* (2.5–4.5 μm in *P. rizhaoensis* vs. 2–2.5 μm in *P. ginkgoana*) [1,24].

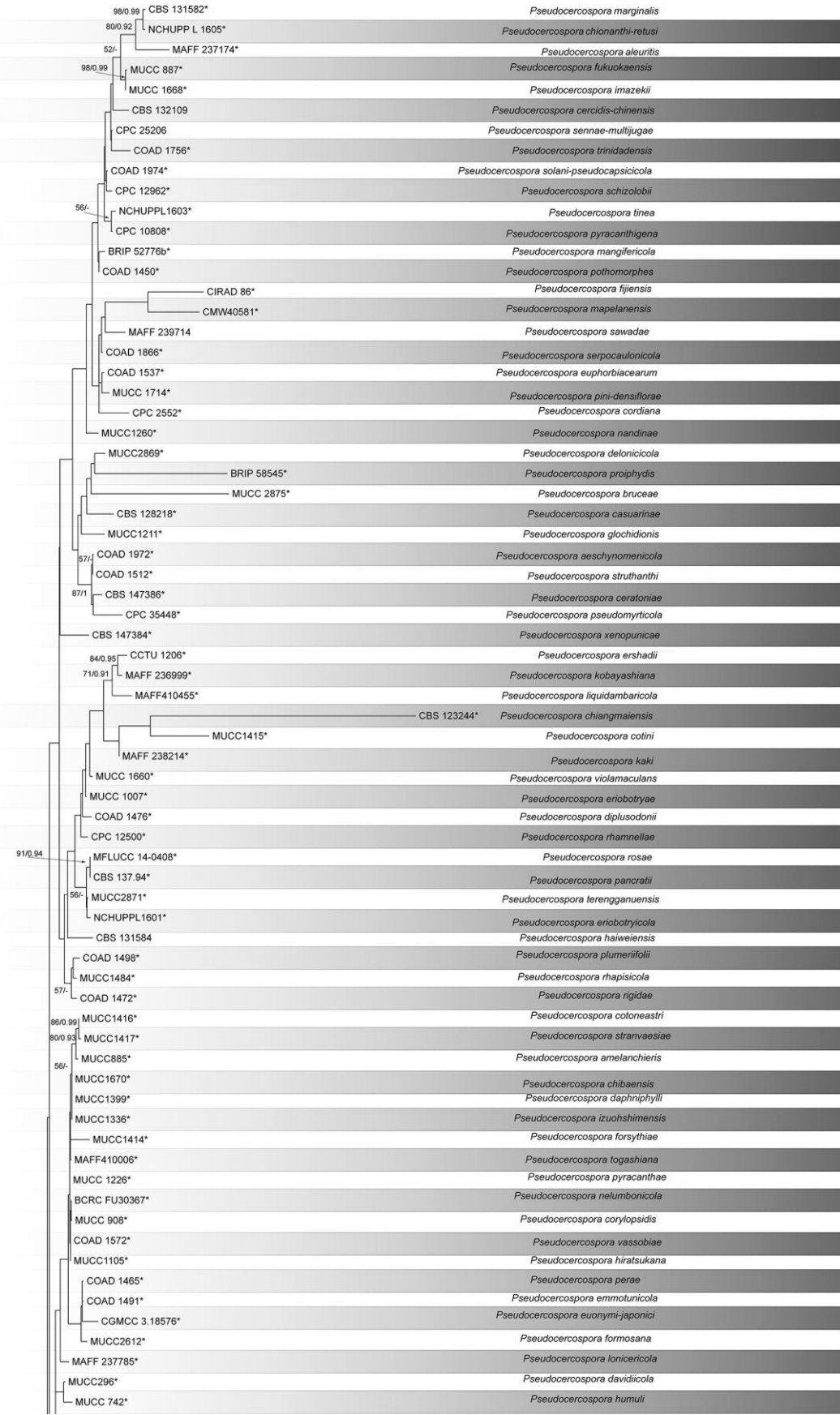

**Figure 2.** *Cont.*

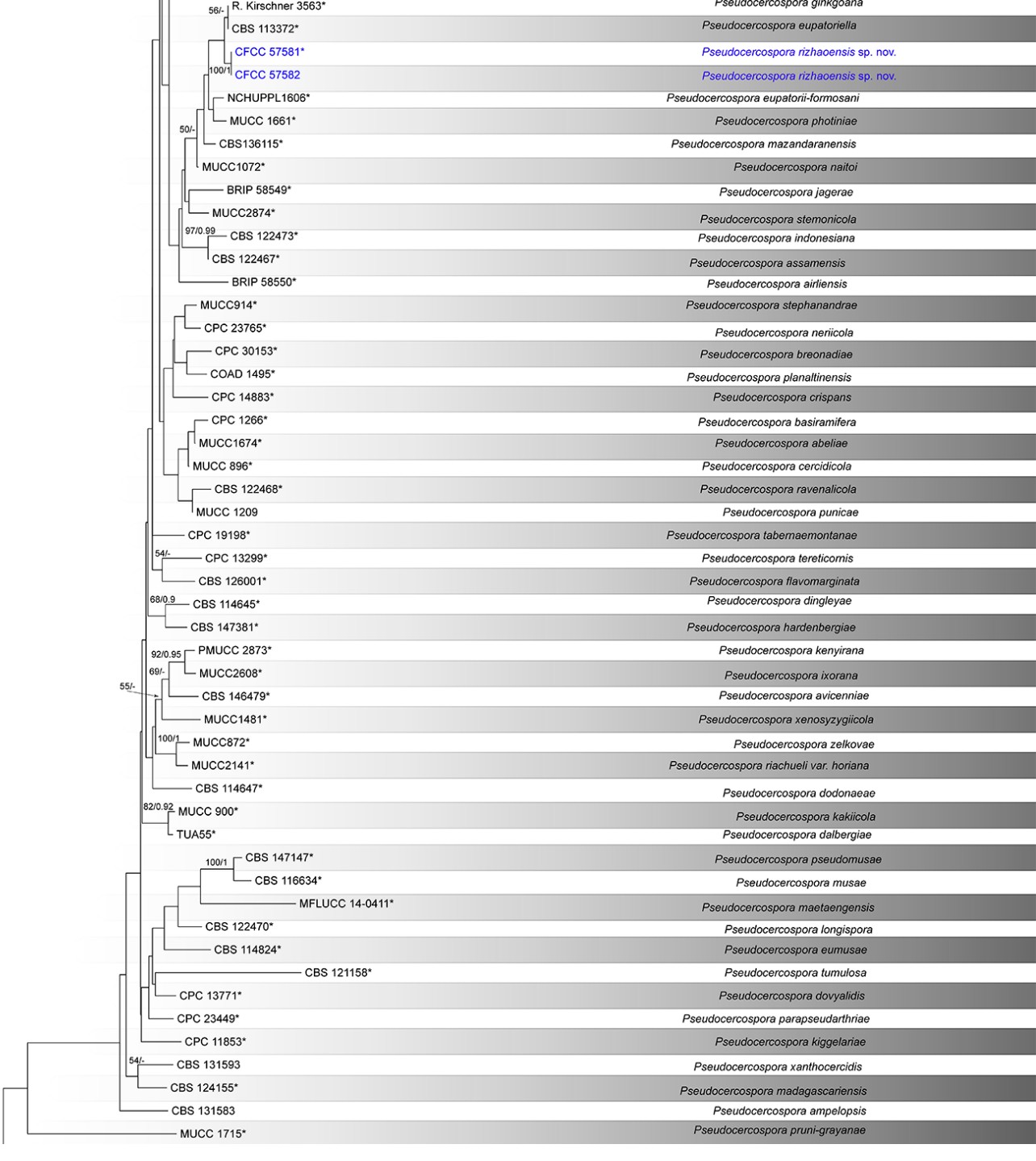

**Figure 2.** *Cont.*

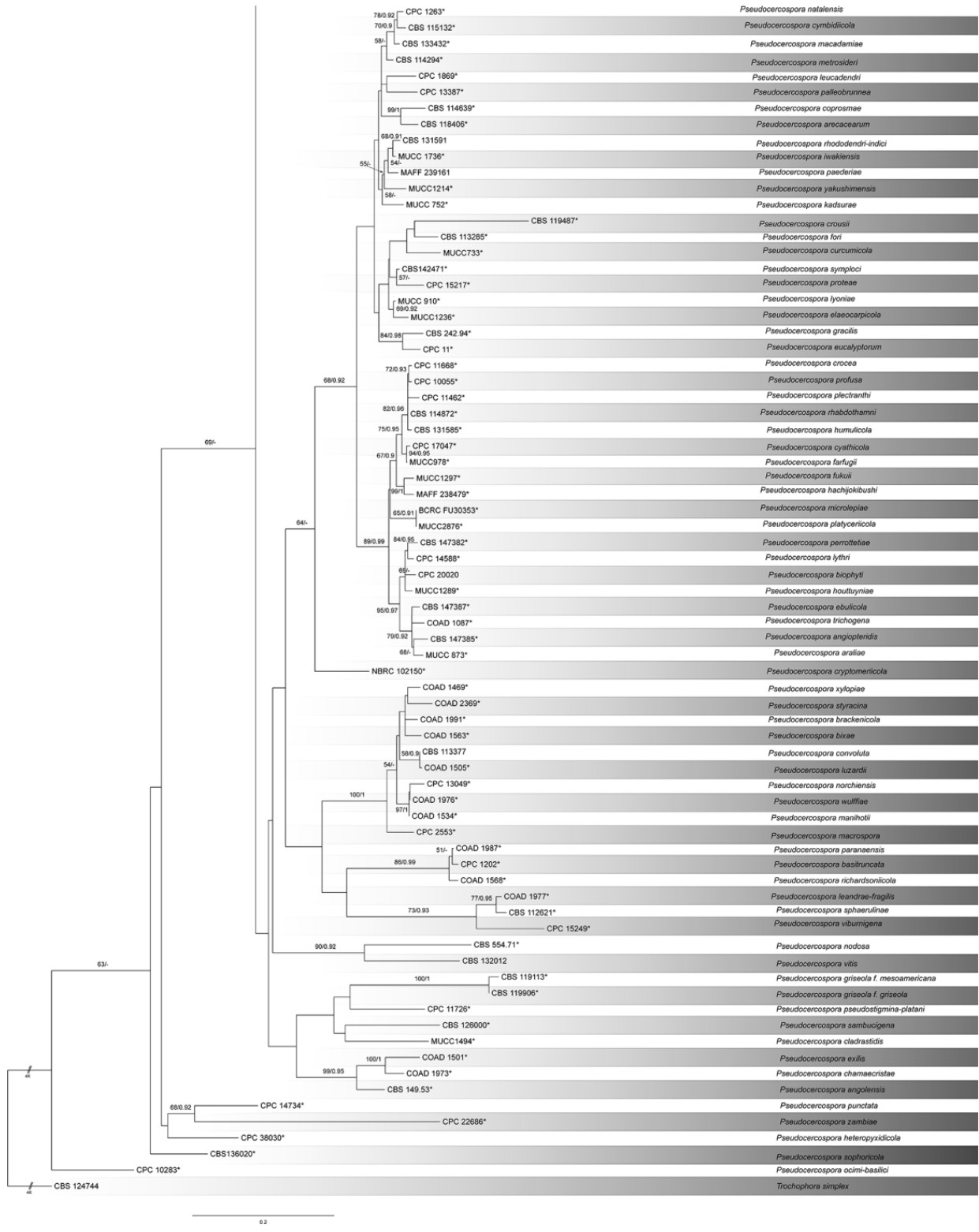

**Figure 2.** Phylogram of *Pseudocercospora* resulting from a maximum likelihood analysis, based on a combined matrix of ITS, LSU, *act*, *rpb2* and *tef1*. Numbers above the branches indicate ML bootstraps (**left**, ML BS ≥ 50%) and Bayesian Posterior Probabilities (**right**, BPP ≥ 0.90). Ex-type strains are marked with *, and the new species proposed in the present study is marked in blue.

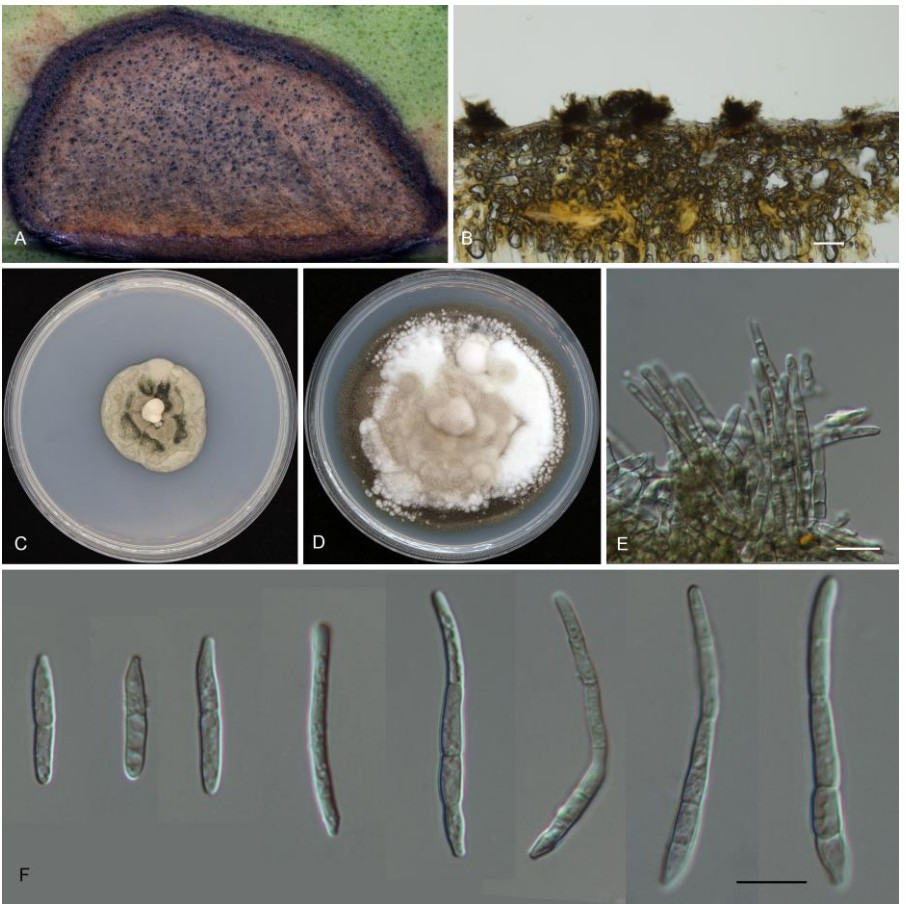

**Figure 3.** Morphology of *Psedocercospora rizhaoensis*. (**A**) Disease symptom on the host leaves; (**B**,**E**) stroma and conidiophores; (**C**) colony on PDA plates at 10 d; (**D**) colony on PDA plates at 20 d; (**F**) conidia. Scale bars: (**B**) = 100 μm; (**E**,**F**) = 10 μm.

*3.3. Pathogenicity Tests*

Similar leaf spot symptoms were reproduced on the *Ligustrum japonicum* leaves after inoculated 20 days, while no symptoms were observed on the control leaves. The respective inoculated fungi were re-isolated from leaves' lesions and were identical to *Ligustrum japonicum* by using morphological characteristics and phylogeny.

**4. Discussion**

The genus *Psedocercospora* was previously considered as an anamorphic state of *Mycosphaerella* or having mycosphaerella-like teleomorphs, but it is now treated as a genus based on phylogeny and morphology [1,25–28]. Now members within this genus are distinguished from each other based on combined approaches of host association, conidia characters and gene sequences [1].

In the pathogenicity test, the leaves after inoculating those isolates showed the same symptoms as disease that occurred in the field, and those isolates could be re-isolated from the lesions. Based on those data, *Psedocercospora rizhaoensis* is considered as the causal agent of the *Ligustrum japonicum* leaf spot disease in China.

*Pseudocercospora ligustri* was recorded causing *Ligustrum japonicum* leaf spots in the USA and *Ligustrum japonicum* 'Howardii' leaf spots in China [29,30]. For this fungus species, no DNA data are available from the type material (IMI 91224 collected in the USA) [29], and three genes, namely ITS, *act* and *tef1*, were sequenced based on the samples collected in China (JS1201, JS1202 and JS1203) [30]. Morphologically, conidiophores of *Psedocercospora rizhaoensis* are narrower than those of *P. ligustri* (2–3.5 μm in *P. rizhaoensis* vs. 3–4.2 μm

in *P. ligustri*) [29,30]. In addition, *P. rizhaoensis* differs from *P. ligustri* in the sequence data (3/470 in ITS, 3/222 in *act* and 1/309 in *tef1*) [30].

As shown in Figure 2, several species were not well-distinguished in the phylogram based on combined loci of ITS, LSU, *act*, *rpb2* and *tef1*. This may be caused by the absence of sequence data for those species (Table 1). Chen et al. [25] demonstrated that ITS is the genus DNA barcode, and the *act*, *rpb2* and *tef1* genes are the species DNA barcode. More phylogenetic analyses to infer the species relationships are necessary in following studies by employing more genes.

**Author Contributions:** Conceptualization, Y.L.; methodology, Y.L. and S.G.; software, S.G.; validation, J.L., X.Y. and Y.L.; formal analysis, Y.L.; investigation, Y.L.; resources, Y.L.; data curation, Y.L.; writing—original draft preparation, Y.L.; writing—review and editing, Y.L.; visualization, Y.L.; supervision, Y.L.; project administration, X.Y.; funding acquisition, X.Y. All authors have read and agreed to the published version of the manuscript.

**Funding:** This research was funded by the Shandong Province Pasture Industry Technology System Project (SDAIT-23-03) and College Youth Science and Technology Support Program of Shandong Province (2021KJ087).

**Institutional Review Board Statement:** Not applicable.

**Data Availability Statement:** Not applicable.

**Conflicts of Interest:** The authors declare no conflict of interest.

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
