# Peer review of "Pseudocercospora rizhaoensis sp. nov. Causing Leaf Spot Disease of Ligustrum japonicum in China"

_diversity, doi:10.3390/d14110990_

Round 1

Reviewer 1 Report

This paper presents new pathogenic species Pseudocercospora rizhaoensis based on molecular and morphological data. Undoubtedly, this is an important finding, however, I have several remarks that need to be addressed before the publication.

Introduction
I guess the introduction must be improved and be more informative, for example:
How many species does the genus include, how many species occur in China, how many known species are sequences, what are the known barcoding genes, and what is the barcoding gap for species elimination in this genus based on a given gene? Why the authors chose exactly these genes for their multigenerational phylogeny, etc?

Methods

I would mention how many leaves or diseased areas were transferred for culture. I guess you sequenced only 2 tubes? Or you had more replicates and only these 2 worked?

I would also make the table and write the used primers pairs besides using the references.

Taxonomic description
I would include the hierarchical information (I.e. order, class, family, subfamily ….)

What about morphological similarities and differences between closely related species? Is it possible to distinguish them if I exclude the molecular and host data?

Mention the barcoding gap between closely related species, the main morphological differences, ecological differences, etc. Can co-infection occur?

Discussion
I suggest adding a few lines explaining what was the advantage of multigene analysis. Did you also run individual trees? If yes, I would suggest it to submit it as supplementary data and say a few words about how individual gene topologies were different from multigen phylogeny and highlight the better resolution of your multigenerational tree.

Author Response

Dear reviewer, 

Your suggestions are raeally helpful for us to improve this manuscript, thanks a lot. 

The details are included in the attachment.

Bests

Reviewer 2 Report

This work provides one new species of Pseudocercospora, causing leaf spots of Ligustrum japonicum in China, based on the multi-locus phylogeny and morphological characters. The paper has been well shaped, and there are still some issues should be addressed to improve it. An annotations text is provided.

Author Response

(The authors gave the same response as above.)

Reviewer 3 Report

The paper is well written and material and methods are correct and accurate. However, there is another recent paper reporting Leaf spots caused by Pseudocercospora species on Ligustrum japonicum in China (First Report of Leaf Spot Caused by Pseudocercospora ligustri on Ligustrum japonicum ‘Howardii’ in China). It seems to me that these species (P. ligustri) are not included in the analysis, both the morphological and DNA-based. Since I think, it could be common in your country, a comparison with your Pseudocercospora should be included and discussed in the Discussion section. In addition, I would suggest to add a little bit more information about the state of art of Pseudocercospora on Ligustrum japonicum in the introduction section.

Title: put “causing” instead of “Causing”

Line 14, abstract: act, tef1 and rpb2 should not be in italic

Line 60: the italic is not needed for act, the same at line 61, 109, 113 and 123 for tef1, rpb2, and at line 65 for (tef1, act and rpb2)

Line 158: In 3.3. Pathogenicity testing, replace “testing” with “tests”

Author Response

(The authors gave the same response as above.)
